# Real-Time Interpolated Rendering of Terrain Point Cloud Data

**DOI:** 10.3390/s23010072

**Published:** 2022-12-21

**Authors:** Jaka Kordež, Matija Marolt, Ciril Bohak

**Affiliations:** 1Faculty of Computer and Information Science, University of Ljubljana, Večna Pot 113, 1000 Ljubljana, Slovenia; 2Visual Computing Center, King Abdullah University of Science and Technology, Thuwal 23955, Saudi Arabia

**Keywords:** point clouds, interpolation methods, terrain rendering

## Abstract

Most real-time terrain point cloud rendering techniques do not address the empty space between the points but rather try to minimize it by changing the way the points are rendered by either rendering them bigger or with more appropriate shapes such as paraboloids. In this work, we propose an alternative approach to point cloud rendering, which addresses the empty space between the points and tries to fill it with appropriate values to achieve the best possible output. The proposed approach runs in real time and outperforms several existing point cloud rendering techniques in terms of speed and render quality.

## 1. Introduction

Three-dimensional acquisition sensors allow us to easily capture diverse shapes into a digital model resulting in a set of points, called a point cloud that lies on the surfaces of objects. Such scanners are used in many different scenarios e.g., object acquisition [1,2], acquisition of environment for self-driving car navigation to improve environmental awareness [3], autonomous planetary exploration [4], or environment acquisition for geographic and geodetic purposes with airborne sensors [5]. The best way to efficiently show these kind of data are by rendering it on screen. This can be achieved using different rendering techniques, but one should aim for an ideal rendering output, which would produce an image indistinguishable from a photograph taken from the same point where the point cloud data were acquired. On the other hand, rendering such data in real-time is often desired, even if this means sacrificing quality.

This is even more true when the data are visualized using limited resources such as on mobile devices or in the web browser. For such purposes, developing dedicated techniques that balances the rendering speed and quality is meaningful. Besides the image quality, rendering time is also crucial. 3D sensors on self-driving cars can produce a new point cloud every 50 milliseconds. Because of other moving objects (nearby traffic, pedestrians, etc.) and the fact that the car itself is moving through space, the new point cloud can be very different from the previous one. If we want to render the data in real time, an efficient algorithm is needed that does not require any time-consuming preprocessing steps.

In our case, we are interested in rendering terrain point cloud data acquired using LiDAR [2] sensors mounted on airplanes while flying over the desired part of the terrain. Such datasets consist of billions of points that store different parameters, such as the intensity of the returned signal, return number (one signal can be reflected from multiple targets), time of acquisition, and other data. For visualization purposes, LiDAR data are usually coupled with color information obtained from the ortho-photo data, which make direct point cloud rendering of terrain data much more meaningful. Due to the amount of data available in LiDAR datasets, the need for fast real-time point cloud rendering techniques is in high demand.

To make the applications accessible to as many users as possible, it is meaningful to implement them using web technologies. In the case of 3D visualization, the most widely adopted standard is Web Graphics Library (WebGL) 2.0 (https://registry.khronos.org/webgl/specs/latest/2.0/, accessed on 19 December 2022), a JavaScript application programming interface, which is available in most browsers for desktop and mobile platforms. WebGL 2.0 allows hardware-accelerated 2D and 3D rendering in browsers without needing extensions and/or plugins and was used to implement our method.

The main contribution of this work is a novel adaptive real-time point cloud rendering technique that identifies which points are on the frontmost surfaces using a depth map and filters out the distant points. We evaluate our method by comparing it with several existing point cloud rendering techniques regarding rendering speed and image quality.

## 2. Related Work

### 2.1. Point Cloud Rendering

The most straightforward way to render point cloud data is to draw points on the canvas using fixed-sized objects. A comparison of such techniques was made by Sainz and Pajarola [6]. These objects can be squares, circles, or any other shape. The only thing we need to consider when doing that is their order, which means starting with the most distant ones and drawing closer ones over them. This approach is straightforward to implement as most graphical libraries already support rendering with fixed-sized objects, and the ordering is already an integral step in the graphical pipeline. Rendering an extremely large point cloud in real time using different dedicated acceleration structures and exploitation of rendering hardware architecture was researched by Schütz et al. [7,8,9,10,11]. While this is an appropriate way to render wast amount of points, the approaches do not address how to fill the empty space between them.

Fixed-sized objects approach can be improved by bending the edges of fixed size away from the camera. The result is a parabolic shape that ensures correct occlusions when points are very close to each other, as was presented by Schütz and Wimmer [12] on 3D scanned point cloud data. If the objects are large enough to fill the empty regions, the result will be the same as with the nearest neighbor algorithm. Furthermore, the researchers have used transparent splats [13] in combination with deferred rendering [14]. One of the ways to fill the gaps between the primitives is to use elliptically weighted averaging [15,16], which also prevents noise in areas with higher point densities.

In its basic form, the nearest neighbor algorithm searches each pixel’s surroundings to find the nearest projected point. The images produced by this technique are called Voronoi diagrams. They never have any holes between the points, but the edges are sharp, and the method does not consider the point’s depth. Implementation can be achieved by following the basic definition, but that turns out to be very inefficient due to intense texture sampling, which is slow on graphics cards. A faster way is using the JumpFlooding algorithm [17] that computes the image in a logarithmic number of steps with respect to the maximal image dimension. A smaller number of steps may even be used if we know that the points are close enough to each other.

To smooth the sharp edges in the Voronoi diagram, the inverse distance weighting interpolation combines the colors of several nearby points. The color of a particular pixel is a weighted sum of all colors in the point cloud. The weights are usually calculated as inverse distances to the points. The distance may also be raised to the power of the parameter *p*. If *p* is a small number, distant points will have similar weight to the near ones, and the picture will be blurry. On the other hand, a very big value of *p* reduces the influence of distant points, and the image becomes similar to the Voronoi diagram.

Unlike inverse distance weighting, natural neighbor interpolation presented by Tsidaev [18] uses only the values of the nearest points around the pixel. To be precise, it takes only those points from which the pixel would remove some area if added to the Voronoi diagram as a new point. The colors are weighted with one of the two defined equations. The Sibson weights are equal to the area removed from each point, and the Laplace weights consider the length of the border in the removed region in combination with the distance to the point.

The above-mentioned space-filling methods give adequate results and are the ones our approach builds upon and aims to improve.

Another way to approach the sparseness of points is to upsample the point clouds. Zhang et al. [19] present a progressive method for point cloud upsampling via differentiable rendering, which addresses the non-uniform point distribution within the point cloud and is capable of learning local and global point features to cope with non-uniform point distribution and outlier removal. Yu et al. [20] present PU-Net, a data-driven point cloud upsampling approach on point patches capable of learning multi-level point features and expanding a set of points using a multi-branch convolution unit implicitly in feature space. These features are split into a new set of features used for upsampling the point set. Li et al. [21] present PU-GAN, a point cloud upsampling approach that uses a generative adversarial network to learn a variety of point distributions in the latent space and uses this information to upsample points point patches of surfaces. Qian et al. [22] present PUGeo-Net, a geometry-centric network for 3D point cloud upsampling, which uses discrete differential geometry and incorporates it into deep learning by learning the first and second fundamental forms for full representation of the local geometry unique up to rigid motion. Li et al. [23] present a method for point cloud upsampling via disentangled refinement, which uses two cascaded sub-networks, a dense generator for coarse but dense surface output and a spatial refiner for further fine-tuning the individual point location using local and global refinement units.

While rendering denser point clouds gives us better results, it also affects the rendering performance. In many cases of rendering aerial point cloud data, we are already tackling hundreds of millions or even billions of points. Since the mentioned methods do not offer real-time point cloud upsampling, this requires extensive preprocessing and affects the rendering performance significantly.

Point cloud reconstruction aims to reconstruct an implicit or explicit surface representation of the data for rendering purposes. Moreover, the point cloud acquisition methods usually return noisy point cloud representations of the acquired data. Researchers have used different approaches to tackle these problems. Mandikal et al. [24] present DensePCR, a deep hierarchical model for predicting point clouds of increasing resolution with architecture that predicts a low-resolution point cloud first and afterward hierarchically increases its resolution by aggregating local and global point features for grid deformation, yielding uniform, dense and accurate point clouds. Tachella et al. citeTachella2019 presented a real-time 3D reconstruction from a single-photon LiDAR data using point cloud denoisers by combining statistical models with highly scalable computational tools from the computer graphics community and demonstrating 3D reconstruction of complex outdoor scenes. Luo and Hu [25] present an approach for learning the underlying manifold of a noisy point cloud from differentiably subsampled points with trivial noise perturbation and their embedded neighborhood feature, aiming to capture intrinsic structures in point clouds. An autoencoder-like network encoding stage learns both local and global feature representations of points. It samples them with low noise using adaptive differentiable pooling operation, and in decoding, infers the underlying manifold by transforming each sampled point along with the embedded features of its neighborhood to a local surface centered around the point. Choe et al. [26] presented a deep point cloud reconstruction network that simultaneously solves point cloud sparseness, irregularity, and noisiness by first using a 3D spars stacked-hourglass network for the initial densification and denoising. Next, it refines the result using transformers to convert discrete voxels into 3D points.

The above methods give good results but have limitations. Most of them are not capable of real-time denoising and reconstruction, and the one which is was only tested on low field-of-view scenes. Moreover, most of the above methods were designed for specific domains and do not guarantee good results for general point cloud data.

An alternative approach is to use indirect techniques for rendering point clouds, which first transform point clouds into an alternative representation, such as mesh geometry of signed distance fields and then render the newly obtained representation. Berger et al. [27] present an overview of surface reconstruction methods up to 2014, which was recently revised by Camuffo et al. [28]. Researchers have developed numerous approaches for point cloud reconstruction and completion by empowering deep learning. Here, we briefly present a few selected works which tackle the point cloud reconstruction and completion problem. Han et al. [29] present a high-resolution shape completion approach that uses deep neural networks for global structure and local geometry inference. Yuan et al. [30] present a point completion network that directly operates on raw point clouds without structural assumption or annotation about the underlying shape and features a decoder design that enables the generation of fine-grained completion using a small number of parameters. Wen et al. [31] present a point cloud completion approach that uses a skip-attention network with hierarchical folding. Williams et al. [32] present a neural kernel fields approach for reconstructing implicit 3D shapes based on a learned kernel ridge regression. The method works on a variety of point clouds representing single objects or big scenes

The above methods achieve great results but are mostly not real-time methods which again requires extensive preprocessing before real-time rendering is possible.

### 2.2. Web-Based Point Cloud Rendering

Several approaches have tackled the possibility of web-based point cloud rendering. Kuder and Žalik [33] have presented a web-based rendering approach for point-based rendering with client-server design, where original LAS data are converted into quadtree representation and a RESTful web server for serving portions of data on different level-of-detail (LoD), mimicking the Web Map Server design. A similar approach was presented by Evans et al. [34], which supports more point cloud formats and utilizes different acceleration structures. Schütz [35] presented an optimized and modular web-based rendering framework that introduced multiple rendering approaches, including paraboloid rendering of massive point cloud data, which does not need a specialized server but includes a data preparation tool for converting the data into an appropriate format for streamed loading of data with LoD support. Discher et al. [36] presented a scalable WebGL-based approach for visualizing massive 3D point clouds using semantics-dependent rendering techniques for the joint rendering of 2D and 3D geodetic data as well as client-side or server-side rendering.

The presented approaches cover diverse use cases but require either additional server-side functionality or extensive preprocessing of data. Our implementation was carried out using RenderCore (https://github.com/UL-FRI-LGM/RenderCore, accessed on 19 December 2022)—a WebGL 2.0 web-based rendering framework with extensions for processing the point cloud data.

## 3. Background

This section briefly introduces the data point cloud data used for testing different rendering methods and commonly used real-time point cloud rendering techniques. We compared the results of our method with the results of these methods to show the advantages and disadvantages of each technique.

### 3.1. Data

The data used in our experiments are publicly available from the Ministry of the Environment and Spatial Planning of Slovenia and consists of multiple datasets. In the presented work, we use LiDAR point cloud data of the Slovenian landscape, orthophoto images of Slovenia, and the digital terrain model.

The LiDAR point cloud dataset of Slovenian landscape (http://gis.arso.gov.si/evode/, accessed on 19 December 2022) was acquired using the RIEGL LMS-Q780 laser scanner, the IGI Aerocontrol Mark II.E 256 Hz IMU system, and the Novatel OEMV-3 GNSS positioning system at altitudes of 1200 to 1400 m above ground. The postprocessing of the acquired data are presented in-depth in the acquisition report [37]. The data were acquired in 2014 and 2015. On average, the dataset contains five points (first return of LiDAR sensor) per m${}^{2}$. The data are georeferenced in both the Gauß-Krüger coordinate system D48/GK and in the geodetic datum 96 coordinate system D96/TM.

Orthophoto images were obtained from *Portal eProstor* [37]—a repository of publicly available Slovenian geodetic data. The images are similarly georeferenced as the LiDAR data. They are available through a WMTS/REST service as small image tiles of 256 × 256 pixels on 17 scale levels and with a resolution of 0.5 × 0.5 m per pixel on the largest scale.

The digital terrain model was also obtained from the database of Slovenian public data (https://ipi.eprostor.gov.si/jgp/menu, accessed on 19 December 2022). The data are stored as a regular grid with a resolution of 1 m, with the corresponding heights with an accuracy of 0.1 m.

### 3.2. Fixed Point Size Rendering

The simplest point cloud rendering method renders points as fixed-sized primitives such as squares, circles, or other planar primitives. Implementation is simple and straightforward since most graphics APIs already include such rendering. This is also true in our case where we used WebGL 2.0, which supports rendering of primitives GL_POINTS, with the ability to set its size through parameter GL_PointSize in the vertex shader. The API draws such points as rectangles, and their overlapping is handled using a depth test built into the graphics pipeline. When we want to render circles, we can discard the fragments outside the desired radius in the fragment shader. Their distance defines the size of the objects we see, and we can implement a similar transformation also in the rendering process. This is why the primitives closer to the viewer are usually rendered larger than the primitives further away. Despite setting the primitive size in accordance with their distance, the problem of non-uniform point cloud density still results in empty regions—holes in the sparsest parts of the point cloud. The following methods try to address these problems in different ways.

### 3.3. Nearest Neighbor Rendering

Pixels in the resulting image in the color of the background are those that no points from point cloud data are mapped onto. One of the most straightforward approaches to approximate its color is by using the color of its nearest neighbor—the point which is close to the pixel in the resulting image. The resulting image of such an approach is called the Voronoi diagram. We can obtain such a result by first rendering the points we have and then iteratively coloring the remaining points in the resulting image with the color of their closest neighbor. Due to the specific hardware architecture of graphical processing units (GPUs), this is not an optimal approach. Such a method requires many texture sampling steps, which makes it inefficient. In the case of rendering an image with resolution 256×256 pixels, this would result in $256^{2}$ = 4,294,967,296 texture samplings. Several approaches tried to address this problem; one of the most commonly used is Jump Flooding [17,38].

The Jump Flooding algorithm builds the Voronoi diagram for an image in $\left\lceil log_{2}\left(d\right)\right\rceil$ steps, where *d* is the maximal image dimension. In the case of the previously mentioned image of resolution 256×256 pixels, this results in 8 steps or $8\times 256^{2}*8$ = 524,288 texture samplings, where the first 8 corresponds to the number of steps (jumps) and the second 8 to the neighboring pixels. Jump length is determined as $2^{k-i}$, where *k* is the number of all steps and *i* to *i*-th step. In the case above, this results in jump lengths of 128, 64, …, 4, 2, and 1. We show how the Jump Flooding algorithm works on a square image of resolution 16×16 pixels in Figure 1.

In WebGL, Jump Flooding can be implemented using multiple rendering passes, where, first, we render the point cloud with one pixel-sized point and set the background as transparent. In the following render passed, we render one step of the Jump Flooding algorithm at a time. We store the coordinates of the nearest neighbors in additional texture after each step. This texture, together with the original point cloud rendering, is passed into the next rendering pass. In the last pass, we generate the Voronoi diagram by rendering the nearest neighbor for every uncolored pixel in the image.

### 3.4. Inverse Distant Weighted Rendering

While nearest neighbor rendering fills all the empty spaces between points, the resulting image is not very precise. The edges between regions are sharp, which makes the results less convincing. Alternatively, the Inverse distance weighting [39] tries to address this by taking into account several neighbors applied to spatial risk distribution in contaminated site assessment. In the case of point cloud rendering, this scenario is mapped to the color of points, and their distribution in the empty space between the points. The smaller the distance, the greater the influence of the neighboring point color is. The color of an arbitrary point in the image is defined as:(1)$$f\left(p\right)=\sum_{i=1}^{n}\frac{f(x_{i})}{{d(p,x_{i})}^{e}},$$
where *n* is number of points, f(x) is point color, d(x,y) is distance between *x* and *y*, and *e* is the weighting parameter. The greater the *e*, the higher the influence of nearby points is. By increasing the *e*, the output image gets closer to the nearest neighbor rendering. The smaller the *e*, the higher the influence of all neighboring points, resulting in a more blurred output image.

### 3.5. Natural Neighbor Rendering

The downside of the Inverse distant weighted rendering is that you need to consider all the points for calculating arbitrary pixel color. This means that rendering denser point cloud regions is substantially more computationally demanding than rendering sparser regions. The Natural neighbors method [40] only uses a few nearest neighbors to calculate the color of a selected pixel. These are the points by which the Voronoi diagram cell surface area becomes affected. The color of a point is calculated as:(2)$$f\left(p\right)=\sum_{i=1}^{n}w\left(x_{i},p\right)*f\left(x_{i}\right),$$
where *n* is number of points in the image, f(x) is point color, and $w_{i}\left(x\right)$ is its weight. An efficient implementation can build on top of Jump Flooding by approximating the portions of surface area the selected pixel would take from each neighboring point in the Voronoi diagram. The approximation can be achieved by sparsely sampling the changed portion of the Voronoi diagram as is shown in Figure 2.

### 3.6. Paraboloid Rendering

None of the above-presented interpolation rendering methods take into account the depth of points and their overlapping. Even when rendering the points with different-sized primitives, both distant and near points are treated similarly in the rendering process, apart from their size, resulting in rendering the points which should be occluded. We can address this problem by rendering with fixed-sized primitives, curving them away from the viewer and thus turning them into curved surfaces. The final result ideally resembles the Voronoi diagram but additionally implicitly takes into account the occlusion of points. One can use different functions to curve back the edges of the primitives, but most commonly, the quadratic function is used, resulting in a parabolic shape–paraboloid. This is one of the most widely used approaches and is also used in the Potree web-based point cloud visualization framework [35].

In the development of our own approach, we identified two main problems with the existing methods:Efficient implementation on the graphical processing unit, andUnwanted rendering of distant points which should have been occluded.

The first problem is best addressed with Inverse distance weighted rendering (see Section 3.4), which can be efficiently implemented. On the other hand, we can almost entirely suppress the calculation errors, which appear as color noise, by selecting appropriate parameters. The Paraboloid rendering method best addresses the second problem (see Section 3.6).

## 4. Methods

In the previous section, we made a general review of the point cloud rendering techniques. We identified the two main challenges when using interpolation for real-time rendering: (1) the removal of occluded points from the scene and (2) the efficient blending of remaining points into a clear image. Based on the experience from observing the implemented methods, we constructed a new algorithm that combines some ideas from the presented methods to achieve better performance. We present the idea and the algorithms in the following sections.

Our proposed method for rendering point clouds comprises three steps:A depth map of occlusion points is prepared;Filtering occluded points;Rendering the points as splats which are combined into the final image,

which are presented in more detail in the following subsections.

### 4.1. Depth Map Calculation

A depth map is an image that contains depth information about the rendered scene. The color of each pixel indicates the distance to the first object in that direction. WebGL automatically produces it with the main result (usually a colored image). We can use it to filter occluded points from the scene by comparing the distance to each point with the depth map value in the same direction. A smaller depth map values mean that something else is in the way, and the point should be removed.

The main challenge when producing a good depth map is using the best shape and size of primitives. Since there is a limited number of points in the point cloud, the surfaces are not completely covered. Points need to be rendered as larger objects to fill the holes between them and produce a depth map with solid surfaces, as shown in Figure 3a. Any surface gap could cause occluded points leakage as the filtering algorithm would not remove them. On the other hand, the primitives also should not be too large as they could cover smaller holes and remove essential details from the image or occlude a whole chain of points as shown in Figure 3b.

The depth map is created with the paraboloid rendering (see Section 3.6) using the default square-shaped primitives. This is a novel way of depth map calculation which is reasonable for point clouds but not for other types of geometry. Only the depth of each fragment is rendered in the depth map. Their size only depends on their distance from the camera, so points farther away from the viewer are represented using smaller squares than the ones closer to the camera. To avoid covering smaller holes, fragment depth is increased towards the edges. Because the increase is based on a quadratic function, the primitives are shaped as paraboloids.

### 4.2. Filtering

The depth map obtained in the previous step is used to filter out the occluded points in the point cloud. The filtering is carried out on points within the rendering frustum in the vertex processing stage, where the occluded points are simply discarded by moving them out of the viewing frustum, thus ignoring them in the fragment processing stage of the rendering.

### 4.3. Rendering with Splats

The third part of the method focuses on incorporating the remaining points into a final image. This can be achieved in many ways using different approaches. Most use interpolation techniques to calculate values of empty pixels based on nearby points. These approaches often require a lot of texture sampling to find those nearby points, which is very inefficient on most GPU architectures. A different approach must be used to achieve our real-time rendering goal.

Our algorithm tries to approximate the inverse distance weighting interpolation and can be described with the following equations: (3)$$\begin{array}{cc} f(p) & =\frac{\sum_{i=1}^{n}w\left(p,x_{i}\right)*f\left(x_{i}\right)}{\sum_{i=1}^{n}w\left(p,x_{i}\right)}, \end{array}$$(4)$$\begin{array}{cc} w(p,x_{i}) & =\frac{1}{{d(p,x_{i})}^{e}}, \end{array}$$
where Equation (Equation 3) is an adaptation of the Inverse distance weighted rendering presented in Section 3.4 with Equation (Equation 1), weighted using Equation (Equation 4).

Instead of searching every pixel’s neighborhood for points, it adds points to the canvas as splats and calculates their weighted averages in the final step. Splats are modeled as square primitives, and their weight is decreasing relative to the distance from the point.

To achieve optimal results, Gaussian probability density is defined as:(5)$$g\left(x\right)=\frac{1}{\sigma \sqrt{2*\pi }}*\exp\left(-\frac{1}{2}*\frac{x^{2}}{\sigma^{2}}\right).$$

It was used instead of an exponential function in Equation (4). The high values around the center produce a sharp image, while values for x≫σ and x≪−σ fill distant holes between points. The variance ($\sigma^{2}$) can be used as a parameter to determine how sharp the result will be. If this parameter is low, the curve will descend quickly, and the final result will be similar to the nearest neighbor interpolation. On the other hand, high values will cause more overlaps of colors and effectively blur the image.

Our method uses a normalized Gaussian curve, where $\frac{1}{\sigma \sqrt{2\pi }}$ is removed from Equation (Equation 5). The weight in the center is consequently always equal to 1 and is not dependent on the variance.

The canvas is represented as a four-dimensional floating-point texture in WebGL. The first three dimensions keep the color information, while the fourth dimension keeps the sum of weights. To ensure correct calculation, the colors are multiplied with the weight before adding them to the canvas. Adding is carried out on the GPU with blending (using the *GL_FUNC_ADD* function). Parameters S=1 and D=1 ensure that all splats are added to the texture unmodified. This part calculates the numerator part of Equation (Equation 1).

The last rendering step calculates the final color of each pixel by simply dividing each color component with the sum of the weights (denominator part of Equation (Equation 1)).

The resulting images from the individual steps of the proposed method are displayed in Figure 4.

## 5. Results

To evaluate our rendering algorithm, we measured its performance in terms of output quality and rendering time. Results were compared to other popular interpolation-based rendering techniques. All tests were conducted on six scenes see Section A.1, always rendered from the same angle. The scenes were taken from the results of the Slovenian national surface LiDAR dataset [37]. Several locations were picked from the dataset, cropped, and filtered to reduce their size. Color information was added using LiDAR-Orthophoto Merging software (https://github.com/jakakordez/lidar-orto-photo, accessed on 19 December 2022).

The test cases were selected so that all the challenges related to point cloud rendering would be covered equally. The scenes contain flat surfaces with complex textures to test texture reconstruction, as well as overlapping objects (e.g., buildings and mountains) to test the rendering of occluded points. The tests were conducted on a consumer-grade computer with an AMD Ryzen 1700 processor, 16 GB of memory, and an NVidia GeForce RTX 2070 graphics card with 8 GB of memory.

### 5.1. Performance Evaluation

As the algorithm was implemented using WebGL 2.0, we used Google Chrome and its developer tools to measure rendering time. We used the performance profiler to measure the execution of the rendering function as well as the GPU utilization. The measured values were averaged over multiple consecutive frames for all six test scenes. The average results are available for each tested algorithm in Table 1.

### 5.2. Qualitative Evaluation

Our test models were obtained by scanning the terrain using LiDAR technology, so we did not have any mesh models or photos for reference. We used a 2D Delaunay triangulation to reconstruct mesh models, render them and use those images as our targets. A simple C# program was used to project the points on a horizontal plane by removing their vertical coordinate. The algorithm from the *DelaunatorSharp* (https://github.com/nol1fe/delaunator-sharp, accessed on 19 December 2022) library connected those points with triangles. In the end, the removed coordinate was returned to the points, and the model was saved as an OBJ file. An example scene rendered with all of the rendering methods is shown together with the close-up in Figure 5. Some additional comparisons are shown in Appendix A, Figure A2.

The similarity between the algorithm output and the reference image was calculated using three metrics:

Learned Perceptual Image Patch Similarity (LPIPS) metric presented by Zhang et al. [41] is used to estimate the perceptual similarity between two images and essentially computes the similarity between the activations of two image patches for some predefined network, and it matches human perception well. A low LPIPS score means that images are more perceptually similar.

Peak Signal-to-Noise Ratio (PSNR) metrics expresses a ratio between the maximum possible value of a signal and the power of distorting noise that affects the representation quality. A high PSNR score means that the two images are more similar.

Structural Similarity Index Measure (SSIM) [42] is a perceptual metric for quantifying image quality degradation caused by processing, data compression, or transmission losses. A high SSIM score means that images are more perceptually similar.

For all six test scenes, we calculated all of the above metrics between the rendering output from a particular method and the target image produced by the mesh rendering method. The average score values for each metric are presented in Table 1. The detailed results for each technique and scene are available in Section A.3.

## 6. Discussion

The performance evaluation shows that our approach is slower than the two fastest methods (rendering with fixed points and paraboloids) and has similar performance to the inverse distance weighted rendering with splats. Still, the method is a real-time rendering method with more than 180 frames-per-second performance on the test scenes. On the other hand, our approach has the best similarity metrics scores on average. While the comparison to the mesh representation of the point cloud data does not perfectly reflect how good the reconstruction is for non-top-down views, it is the best approximation we could obtain using the available data. The images show that our method fills most of the holes in the point clouds, resulting in a better-reconstructed scene.

Despite promising results, our method could use further improvements. The most noticeable flaw is observed when the camera moves around the scene. The depth image that the filtering part of the algorithm uses can still have smaller holes. Consequently, some points from the background may become visible for a short time as the camera moves. This causes a flickering effect on the screen, which should be addressed in the future. Similar flickering is also present in some of the other presented methods.

The presented method is especially meaningful when the point cloud data are less flat (e.g., in mountains and residential areas). In these scenes, our method outperforms others even more than in flat scenes.

One downside of our method is that some regions in the resulting images are blurrier than with other methods; the effect is more apparent in scenes with smaller objects and lower densities of point clouds. This could be solved to some degree with the use of a post-processing sharpening filter.

While newer deep-learning-based methods, which were not assessed in this paper, undoubtedly produce better results, the inference step is still far from real-time. In addition, until new hardware and deep model optimization enable us to run inference of such model in real time, the presented method is still very relevant for point cloud rendering.

## 7. Conclusions

During this study, we managed to develop our own algorithm for real-time point cloud rendering. The algorithm is able to filter occluded points from the scene and fill empty spaces on the canvas with colors from nearby points. Other interpolation-based algorithms were also implemented for performance and qualitative comparison. All algorithms were evaluated on a set of six test scenes. Reference renderings were also made from mesh reconstructions of those models. The results show that the presented point cloud rendering method is capable of real-time rendering of point cloud data and outperforms other popular real-time or interactive point cloud rendering methods.

The method could be further improved with a better filtering step, which would use some kind of density map. A density map is an image of the scene with higher pixel values where points are projected closer one to another. Such a map can be efficiently generated using the same technique as the color blending part of our method. If the render pass that is generating the depth image would use this density map to increase the size of paraboloids where density is low, it could potentially fill all empty spaces and therefore give better results. Additionally, when the computational power of new hardware will enable real-time inference of deep models, the method could be adapted in such a way that the filtered points would be sent to a deep model, which would then infer the output image. An additional problem that might arise is how to address the temporal coherence of such models throughout the sequential frames.

## Figures and Tables

**Figure 1 sensors-23-00072-f001:**
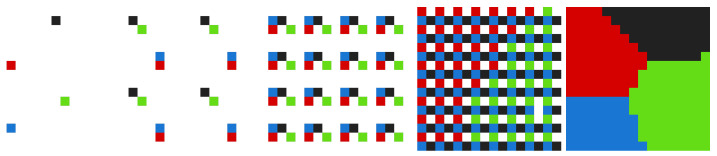
From (**left**) to (**right**): consecutive steps of Jump Flooding algorithm for image of resolution 16×16 pixels.

**Figure 2 sensors-23-00072-f002:**
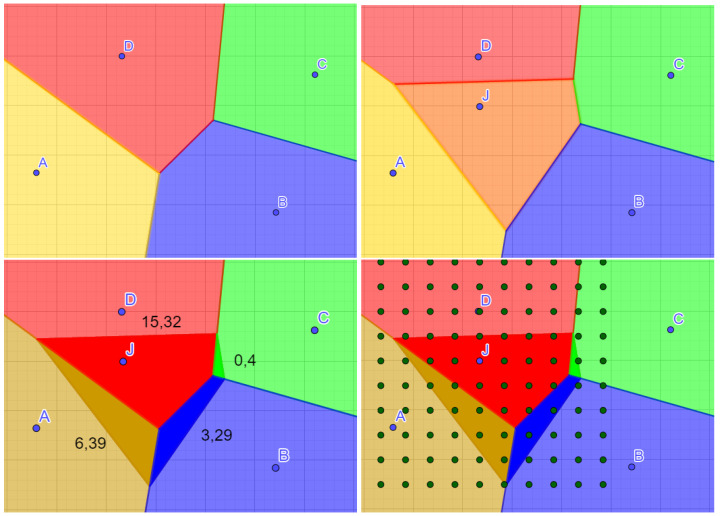
(**Top left**) original Voronoi diagram; (**top right**) updated Voronoi diagram with a new point; (**bottom left**) surface area portions taken by the new point; and (**bottom right**) sparse sampling of the affected part of the Voronoi diagram for estimating the surface area portions.

**Figure 3 sensors-23-00072-f003:**
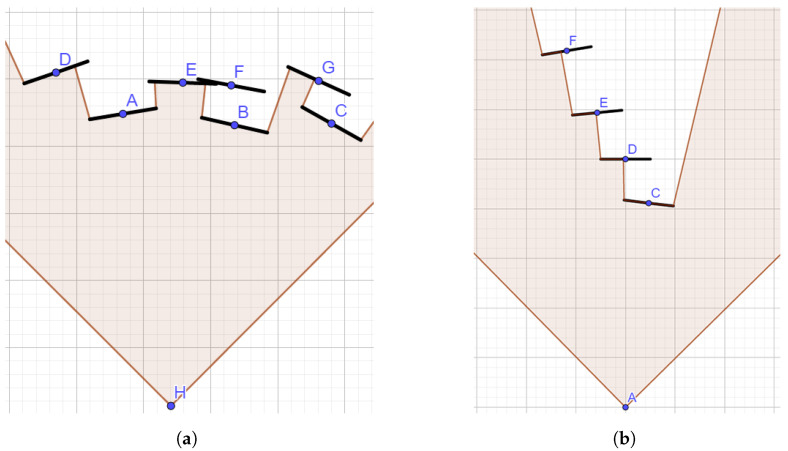
Point filtering with use of depth map. (**a**) Points F and G are filtered out, since they are occluded by points B in C; (**b**) a side view on the surface defined with points C–F. Point C occludes point D, which in turn occludes point E, which in turn occludes point F.

**Figure 4 sensors-23-00072-f004:**
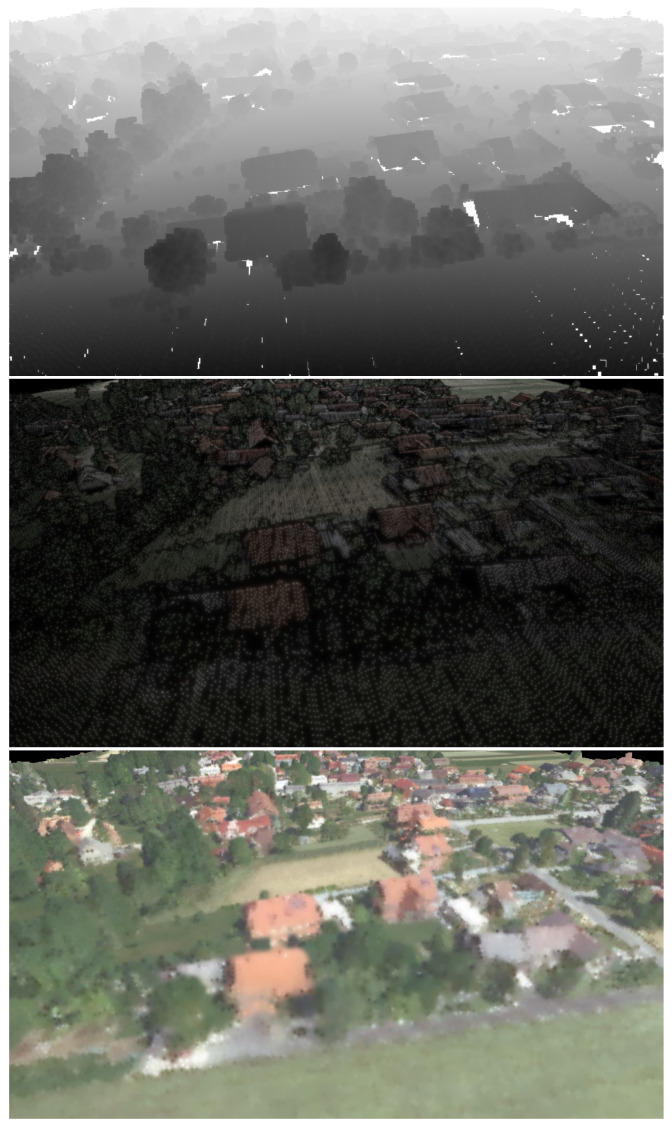
From left to right: step 1—depth map of occlusion points, step 2—filtering occluded points, step 3—final rendering using splatting.

**Figure 5 sensors-23-00072-f005:**
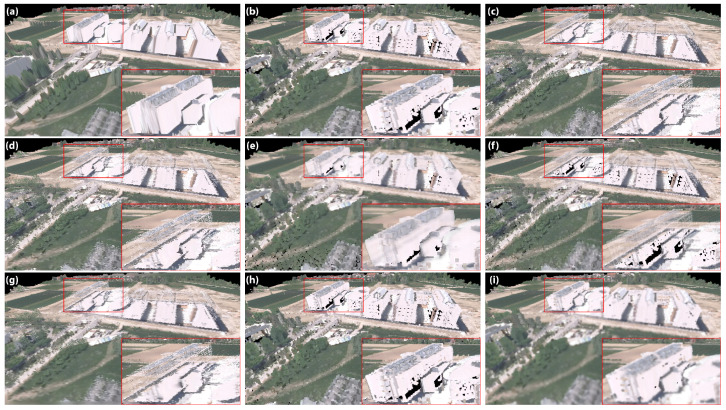
Rendering results for a scene with complex buildings of the presented rendering methods: (**a**) mesh; (**b**) fixed points; (**c**) nearest neighbor; (**d**) jump flooding; (**e**) inverse distance weighted; (**f**) inverse distance weighted with splats; (**g**) natural neighbor; (**h**) paraboloids, and (**i**) our approach.

**Table 1 sensors-23-00072-t001:** Results of all implemented algorithms with values averaged across all test cases. In bold are the values of best values according to the used metric.

Approach	Performance		Quality		
	Time ↓	GPU ↓	LPIPS ↓	PSNR ↑	SSIM ↑
Rendering with fixed points	**0.2 ms**	**0.2 ms**	0.135	18.452	0.491
Nearest neighbors interpolation	11 ms	9.1 ms	0.165	18.290	0.526
Jump Flooding algorithm	132 ms	108 ms	0.164	18.256	0.524
Inverse distance weighting	233 ms	232 ms	0.145	19.033	0.495
Inverse distance weighting with splats	5.2 ms	2.7 ms	0.233	17.344	0.438
Natural neighbors interpolation	188 ms	161 ms	0.137	18.935	0.577
Rendering with paraboloids	0.3 ms	**0.2 ms**	0.139	18.541	0.497
Our approach	5.3 ms	3.0 ms	**0.091**	**21.475**	**0.655**

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
