# Peer review of "Real-Time Interpolated Rendering of Terrain Point Cloud Data"

_sensors, 2022, doi:10.3390/s23010072_

Round 1

Reviewer 1 Report

This paper studied the interpolated problem in point cloud rendering, and proposes a real-time point cloud rendering technique. The results were compared to other popular interpolation-based rendering techniques. I found the paper to be overall well written and much of it to be well described, but some of the description of the paper to be too detailed, while the description of some very important points was inadequate or completely missing. The significance of this paper is not expounded sufficiently. The authors need to highlight this paper's innovative contributions. I explain my concerns in more detail below.

Section 1, since the algorithm was implemented using WebGL 2.0, the author should introduce it more.

Section 3, it's not clear what has been utilized from the existing work and what is new (proposed by the authors). It should be clearly discussed. In the introduction, the author introduced that the main contribution of this work is a novel adaptive real-time point cloud rendering technique that identifies which points are on the frontmost surfaces using a depth map and filters out the distant points., but in the third section (Methods), what the author focuses on is the interpolation method which tries to approximate the inverse distance weighting interpolation. And the contribution to the depth map calculation is not clear enough. The author would better list a formula to describe his own depth map calculation method in detail.

Section 4, It can be seen that the author has done a lot of work, eight algorithms were used, all tests were conducted on six scenes, always rendered from the same angle. However, the author did not describe the basis for selecting the scenes, nor did he show the selected scenes. It is suggested to add a discussion on the adaptability of different methods in different types of scenes.

Reviewer 2 Report

The paper deals with an interesting theme highly topical in the context of the current World research trends. The authors in this work propose an alternative approach to point cloud rendering, which addresses the empty space between the points and tries to fill it with appropriate values to achieve the best possible output. The proposed approach runs in real time and outperforms several existing point cloud rendering techniques in terms of speed and render quality. I appreciate especially the detailed map outputs, as well as the interpretation of the obtained results. In order to meet the objective of the paper, the author chose an adequate methodical apparatus based on the use of historical data, orthophotos, LIDAR data, digital terrrain model and modern geoinformatics equipment with using rendering methods.

The title of the paper is acceptable and adequate and no changes are necessary. I find the abstract acceptable and well structured. The manuscript has a sufficient scientific value and the information provided represents widening of knowledge. The conclusions are based entirely on the results and the methods used are adequate. The relation between the scientific value and the extent is acceptable. The language and style of the text are at an acceptable level. The tables and illustrations used in the paper are adequate; however I consider the number of references incomplete. The topic dealt with in the paper is also covered by other authors in papers. I have no other remarks of a rather significant nature concerning the paper. The results are valuable and the scientific paper brings new original data. The manuscript is acceptable after minor revision with minor amendments required; no re-review is necessary. I recommend the paper for the print.

So no elements that should be corrected:

Conclusion: any limitation of your research? So please add it.

I recommend amending the references. This issue is also covered by the newer papers from other authors. I recommend adding some papers into the references.

Figure 2 -  text is very small /little poor quality (especially text inside of pic).

As you see, there is not too much to correct according to my opinion.

Good luck in your future scientific work.
